# Revolutionizing Pathology with Artificial Intelligence: Innovations in Immunohistochemistry

**DOI:** 10.3390/jpm14070693

**Published:** 2024-06-27

**Authors:** Diana Gina Poalelungi, Anca Iulia Neagu, Ana Fulga, Marius Neagu, Dana Tutunaru, Aurel Nechita, Iuliu Fulga

**Affiliations:** 1Faculty of Medicine and Pharmacy, Dunarea de Jos University of Galati, 35 AI Cuza St., 800010 Galati, Romania; dianapoalelungi10@gmail.com (D.G.P.); mariusneagu87@gmail.com (M.N.); dana_tutunaru_cmgl@yahoo.com (D.T.); nechitaaurel@yahoo.com (A.N.); fulgaiuliu@yahoo.com (I.F.); 2Saint Apostle Andrew Emergency County Clinical Hospital, 177 Brailei St., 800578 Galati, Romania; 3Saint John Clinical Emergency Hospital for Children, 800487 Galati, Romania

**Keywords:** artificial intelligence, computer-assisted image analysis, computer-aided diagnosis, pathology, digital pathology, immunohistochemistry

## Abstract

Artificial intelligence (AI) is a reality of our times, and it has been successfully implemented in all fields, including medicine. As a relatively new domain, all efforts are directed towards creating algorithms applicable in most medical specialties. Pathology, as one of the most important areas of interest for precision medicine, has received significant attention in the development and implementation of AI algorithms. This focus is especially important for achieving accurate diagnoses. Moreover, immunohistochemistry (IHC) serves as a complementary diagnostic tool in pathology. It can be further augmented through the application of deep learning (DL) and machine learning (ML) algorithms for assessing and analyzing immunohistochemical markers. Such advancements can aid in delineating targeted therapeutic approaches and prognostic stratification. This article explores the applications and integration of various AI software programs and platforms used in immunohistochemical analysis. It concludes by highlighting the application of these technologies to pathologies such as breast, prostate, lung, melanocytic proliferations, and hematologic conditions. Additionally, it underscores the necessity for further innovative diagnostic algorithms to assist physicians in the diagnostic process.

## 1. Introduction

Immunohistochemistry (IHC) is a fundamental component of pathology, providing indispensable insights into the molecular complexities of various diseases. This technique utilizes antibodies to identify specific antigens within tissue samples. The chromogen 3,3′-Diaminobenzidine (DAB) is widely used to visualize those antigens of interest [1]. Moreover, IHC allows pathologists to identify cellular markers suggestive of specific diseases, like cancer subtypes or infectious agents, contributing to diagnosis, prognosis, and treatment planning across multiple medical specialties.

Digital Pathology (DP) combines the acquisition, management, sharing, and interpretation of pathological data, including slides, in a digital environment. The image acquisition is performed using whole-slide imaging (WSI) scanners. It has been nearly 20 years since the commercial introduction of WSI scanners and, over this period, the development of various WSI devices capable of digitizing entire glass slides has significantly transformed the field of pathology [2]. Images produced by those devices are a rich source of information, exhibiting greater complexity than many other imaging modalities due to their large size (commonly at a resolution of 100 k × 100 k), the inclusion of color information (such as hematoxylin and eosin staining and IHC), the availability of information at multiple magnifications (e.g., ×4, ×20), and multiple z-stack levels (each slice has a finite thickness and generates different images depending on the plane of focus) [3].

After acquiring digital images, various computer applications leveraging AI can be employed to analyze the information contained within the images. For instance, computer-assisted image analysis (CAIA) has been employed to quantify immunohistochemical stains, such as Estrogen Receptor (ER), Progesterone Receptor (PR), and Human Epidermal Growth Factor Receptor 2 (HER-2/neu) breast biomarkers, providing a standardized method for all pathologists to score IHC findings in breast cancer cases [4]. Furthermore, eXtreme Gradient Boosting (XGBoost), an increasingly prominent machine learning (ML) method, utilizes tree boosting to scale effectively and its performance has attracted significant attention from researchers due to its scalability and superior performance, surpassing numerous traditional classifiers in the field of ML [5].

Table 1 provides an overview of the definitions of terms related to AI and their integration within the pathology sector. The authors employed a visual representation, in the form of Figure 1, to illustrate the various tissues with reported AI applications in pathology.

## 2. Role of Artificial Intelligence in Immunohistochemistry

### 2.1. Automated Image Analysis

AI algorithms, particularly those based on computer vision and DL, can analyze IHC-stained digital slides with high accuracy [27]. For instance, a 2020 study by Fassler D.J. et al. [8] illustrated how deep learning-based techniques for brightfield-acquired multiplex IHC can effectively classify and quantify six or more cell types within a single tissue section. Regarding fibroblast growth factor receptor-2 (FGFR2), an emerging IHC marker, which is a major transducer of signals between tumor and its microenvironment (TME), mediating interactions between TME and hormone receptor-dependent pathways [28,29,30,31,32,33,34,35], Braun M. et al. [36] demonstrated the application of an ML algorithm for evaluating this marker’s expression in breast cancer. The study found a significant level of agreement between AI-based assessments and manual evaluations, showing a strong correlation between the two methods.

### 2.2. Integration of AI Software with Digital Pathology

Over the past few years, leading software companies specializing in pathology applications have made notable advancements in the field of IHC. For example, Aiosyn has recently expanded its AI-powered quality control solution (AiosynQC) to support IHC slides in addition to its existing compatibility with H&E slides [37]. This update is designed to enhance DP workflows by automatically detecting and flagging common artifacts in histology slide images, such as air bubbles and areas that are out of focus.

### 2.3. Quantitative Analysis

Recent studies have highlighted the significant role of artificial AI in quantitative analysis within IHC, enhancing the precision and efficiency of diagnostic processes. Specifically, Mindpeak (Hamburg, Germany) has significantly advanced breast cancer diagnostics with its suite of automated image analysis software modules [38]. These modules, including Mindpeak Breast HER2 ROI, Mindpeak Breast Ki-67 HS, and Mindpeak Breast ER/PR, facilitate the automated analysis of digital pathology images of invasive breast carcinoma tissue samples [38].

### 2.4. Standardization and Reproductibility

A study on the AI-assisted interpretation of Ki-67 expression in breast cancer demonstrated that AI models significantly enhance the repeatability and accuracy of IHC assessments [39]. The study involved nine pathologists (juniors, intermediates, and seniors) and used AI to interpret Ki-67 immunohistochemical sections, showing high consistency with the gold standard, thus improving the reliability of diagnostic outcomes [39].

## 3. Literature Review

### 3.1. Methodology

We conducted a comprehensive review of the current literature including original articles that studied various applications of AI in IHC. We performed extensive searches on the Google Scholar, PubMed, and ScienceDirect databases to identify relevant manuscripts. As keywords, we used “artificial intelligence”, “computer-aided diagnosis”, and “computer assisted image analysis”, combined with “digital pathology”, and “immunohistochemistry”. We restricted our search to papers published in English between 2014 and 2024 and found more than 200 relevant manuscripts. The inclusion criteria targeted studies that investigated the use of AI in IHC as part of the final histopathological diagnosis. We excluded articles that focus on the implementation of AI algorithms on H&E slides, molecular biology, and editorial comments.

### 3.2. Results

After a thorough review and assessment of the eighty-five articles, we identified and included a subset of nineteen papers that were directly relevant to our research, including eleven on breast cancer, one on prostate cancer, one on lung cancer, two on malignant melanoma, one on cancers of unknown primary origin (CUPs), and three on lymphoid neoplasms. The selected studies offered valuable insights into the application and impact of AI in final histopathological diagnosis using IHC, serving as the foundation for our review.

#### 3.2.1. Breast Cancer (BC)

Breast cancer refers to the erratic growth and proliferation of cells that originate in the breast tissue [40]. According to the latest report from the World Health Organization, breast cancer has become the most common malignant tumor [41]. The most prevalent type of breast cancer histology is invasive ductal carcinoma, affecting 50–75% of patients [42]. This is followed by invasive lobular carcinoma, which occurs in 5–15% of cases [42]. The remaining cases consist of mixed ductal/lobular carcinomas and various other less common subtypes [42]. Another classification of breast cancers is based on molecular subtypes [43]. These are categorized into five groups: luminal A, luminal B, HER2-enriched (human epidermal growth factor receptor type 2), basal-like, and normal breast-like types [43]. The molecular classification based on the expression of ER, PR, and HER2 is straightforward [44]. These three biomarkers are routinely reported by pathology departments, with well-established immunohistochemical staining and reporting protocols, and available quality control programs in many centers [44]. Using the Ki-67 proliferation index along with HER2 expression facilitates the differentiation between luminal A and luminal B subtypes [45]. It has significant implications for personalized medicine to determine the subtypes, which exhibit distinct symptom characteristics in terms of metastasis, recurrence, and sensitivity to different treatments [46].

Breast carcinoma originates from the mammary epithelium and initially causes a premalignant proliferation within the ducts, known as carcinoma in situ (CIS) [47]. However, the cancer cells can eventually acquire the ability to penetrate the basal membrane and invade the surrounding tissues [47]. The distinction between benign, in situ, and malignant proliferations is determined using immunohistochemical markers such as p63, SMA (smooth muscle actin), and cytokeratin (CK) 5/6 [48]. These markers highlight the presence of myoepithelial cells, which are typically retained in benign and in situ lesions but are absent in invasive carcinomas [48]. After extensive research, we did not find articles that specifically address the use of AI algorithms for the detection of basal/myoepithelial markers on IHC slides. However, there are several algorithms developed for H&E slides, such as those used for tumor segmentation and classification, which could potentially be adapted for this purpose. For example, the GALEN algorithm can analyze entire core needle biopsy WSIs and detect various types of breast lesions, including invasive and in situ carcinoma, as well as non-obligate precursors such as atypical hyperplasia [49,50,51,52]. Additionally, it can identify benign findings such as sclerosing adenosis, fibroadenoma, and fat necrosis [49,50,51,52].

The evaluation of hormone receptor (HR) status serves as both a prognostic and predictive factor in breast cancer (BC), making it an essential step in tailoring therapy for BC patients [53,54,55]. Numerous studies have explored the use of AI for detecting ER and PR on IHC slides. Specifically, Rawat R. R. et al. reviewed 95 articles and discovered that DL algorithms developed to quantify ER and PR expression have shown a correlation exceeding 95% between manual and algorithmic quantification [56]. These findings highlight the robustness of the methodology and demonstrate that AI-aided detection of IHC markers can be as accurate as human assessment [56]. Similar results for ER and PR scoring using IHC-stained images with a deep neural network, comprising an encoder, decoder, and scoring layer, have shown excellent performance, potentially reducing human error and aiding early BC detection [57], although caution is advised for faint staining or new ER-low sub-classes [58], due to the risk of false negative results [59].

The expression of HER2 protein is crucial for making therapeutic decisions in breast cancer treatment [60]. Approximately 15–20% of newly diagnosed invasive breast carcinomas express HER2 oncogene, which is linked to increased tumor progression and metastasis [61,62,63]. The conventional diagnostic method typically classifies HER2 IHC into negative (0 and 1+), equivocal (2+), and positive (3+), based on the intensity of HER2 membranous staining and the percentage of tumor cells that show this staining [64]. However, according to the recently published DESTINY-Breast 04 trial results, HER2-low tumors were defined as having a score of 1+ on IHC or 2+ on IHC with a negative in situ hybridization (ISH) result [65]. This cohort of patients demonstrated significantly longer progression-free and overall survival when treated with trastuzumab deruxtecan compared to chemotherapy [65]. Several studies have investigated the use of AI in determining HER2 status, employing methods such as tumor cell segmentation and the evaluation of HER2 membrane staining intensity and patterns [66,67,68]. For example, Holten-Rossing H. and colleagues [67], using the digital image analysis tool DIA HER2-CONNECT, demonstrated that automated DIA assessment increased both sensitivity and specificity to 100% and 95.5%, respectively. In comparison, manual assessment showed a sensitivity of 85.0% and a specificity of 86.0% [67].

Ki67 is an immunohistochemical nuclear marker widely used in surgical pathology, where nuclear immunoreactivity indicates cell cycling from the G1 to the S phase, and the percentage of Ki67-positive tumor cells (Ki67 index) provides an estimate of the tumor’s growth fraction [69]. In BC, Ki67 is used as a prognostic tool, and one of the first and most widely adopted AI algorithms was Ki67 proliferation index scoring, provided by many freely available platforms [39,70,71,72]. Specifically, Li L. et al. demonstrated in their study that AI counting for Ki-67 is highly consistent with the gold standard, meeting and even surpassing the International Breast Cancer Working Group’s recommended cell number range [39].

Tumors consist of not just cancerous cells but also a variety of non-malignant cells, including those from the immune, vascular, and lymphatic systems, as well as fibroblasts, pericytes, the extracellular matrix, and adipocytes, and those cells can sometimes make up over 50% of the tumor’s composition [73]. Specific killing lymphocytes in the tumor microenvironment (TME) are called tumor-infiltrating lymphocytes (TILs), but their tumor-killing ability is inhibited by immunosuppressive factors in the tumor microenvironment [74]. Building on this approach, scientists used in vitro culture methods to enrich tumor tissue lymphocytes and then transfused them back into the patient, resulting in an anti-tumor effect [75]. Unlike other cellular immunotherapies, TILs are derived from the patient’s own cells without genetic modification and have a specific tumor cell-killing capability [74,76]. Analyzing specific immune cells’ spatial relationships before, during, or after therapy has significant prognostic potential, achievable with commercial and open-source image analysis tools for area-based quantification of immune cells via IHC or immunofluorescence (IF) [77,78,79,80]. Studies have proposed TIL quantification using convolutional networks on image-based IHC-stained sections in gastric, breast, prostate, and colon cancers [81,82,83]. These investigations rely on detecting IHC markers such as CD3 and CD8 for TIL quantification [84]. For instance, Swiderska-Chadaj Z. et al. demonstrated that DL techniques can effectively detect positively stained TILs in IHC, showing significant promise for immuno-oncology [81]. The ability to reliably quantify these cells paves the way for research linking immune cell quantities to tumor progression and treatment response [81].

#### 3.2.2. Prostate Cancer (PC)

Benign prostatic hyperplasia (BPH) followed by prostatic adenocarcinoma constitute the predominant cases of prostatic pathology [85]. Prostate cancer is the second most frequent cancer diagnosis affecting men and the fifth leading cause of death worldwide [86]. Prostate adenocarcinoma, originating from glandular epithelial cells, comprises over 95% of prostate cancer cases, and is diagnosed via histological examination of tissue obtained from transrectal ultrasound-guided needle biopsy [87]. The distinction between benign and invasive glands relies on architectural and cytological features [87]. As a number of benign mimics of PCa and conversely deceptively bland variants of PC exist, IHC is often required [88,89]. The IHC panel recommended by the International Society of Urologic Pathology include CK5/6, 34BE12, P63, and alpha-methylacyl-CoA racemase (AMACR) [90,91].

Paige Prostate is a machine learning algorithm trained on the digital slide archive of Memorial Sloan Kettering Cancer Center (MSKCC) in New York that takes a H&E whole-slide image as its input and classifies the image as “suspicious” for prostatic adenocarcinoma if it detects adenocarcinoma or glandular atypia (such as focal glandular atypia (FGA), high-grade prostatic intraepithelial neoplasia with adjacent atypical glands (PIN-ATYP), or atypical small acinar proliferation (ASAP)); otherwise, it categorizes it as “not suspicious” for prostatic adenocarcinoma if none of these lesions are detected [92]. Although not currently employed for IHC detection of PC, Raciti P. et al.’s study illustrated that in the absence of Paige Prostate Alpha, pathologists exhibited an average sensitivity of 74% and an average specificity of 97% [92]. However, when utilizing Paige Prostate Alpha, the average sensitivity among pathologists notably rose to 90%, with no statistically significant alteration in specificity [92].

The quantification of Ki67 expression by conventional bright-field IHC has been proven to be a strong prognostic parameter in prostate cancer [93,94,95,96,97]. Blessin, N.C. et al. stated that Ki67 can rapidly and reproducibly be analyzed by AI-supported multiplex fluorescence IHC (mfIHC), with the major advantage of strict limitation of the analysis to tumor cells, which cannot be achieved in ribonucleic acid (RNA) or deoxyribonucleic acid (DNA) based panel analyses [98].

#### 3.2.3. Lung Cancer (LC)

Over the past century, lung carcinoma has evolved from a rare and poorly understood condition to the most prevalent cancer globally and the leading cause of cancer-related mortality [99]. Lung cancer is a heterogeneous disease, which comprises various subtypes that hold significance both pathologically and clinically [100,101,102]. These subtypes are categorized into two primary groups based on their main histotype, which carry prognostic and therapeutic implications: small-cell carcinoma (SCLC), accounting for 13% of cases, and non-small-cell carcinoma (NSCLC), comprising 83% of cases [102,103]. Furthermore, the emergence of molecular profiling and targeted therapy has reignited interest in further categorizing NSCLC into adenocarcinoma (ADC) and its variations, squamous cell carcinoma (SqCC), and large-cell lung carcinoma (LCLC) [104,105,106].

In their study, Kriegsmann M. et al. utilized CNNs on histological images stained with H&E, to evaluate their capacity to classify the primary lung cancer subtypes, namely SCLC, ADC, and SqCC [107]. The optimized InceptionV3 CNN architecture achieved the highest classification accuracy and was deployed on the test set. Following stringent Quality Control, image patch and patient-based CNN classification results reached 95% and 100%, respectively, in the test set [106]. Misclassifications primarily involved ADC and SqCC [107].

IHC is employed in lung cancer for various purposes, such as (i) distinguishing between ADC and SqCC; (ii) detecting neuroendocrine markers; (iii) identifying driver genetic alterations (ALK, ROS1, and EGFR); (iv) assessing PD-L1 (CD274) expression; (v) discriminating between lung carcinoma and malignant mesothelioma; and (vi) diagnosing NUT carcinoma [108].

The clinical effectiveness of agents inhibiting CTLA-4 (cytotoxic T lymphocyte-associated protein 4, CD152) and PD-1/PD-L1 checkpoints (programmed cell death protein 1, CD279; programmed death-ligand 1, CD274) has led to swift regulatory approval for treating patients with various solid tumors and hematologic malignancies [109]. The assessment of PD-L1 expression with IHC has emerged as an important predictive biomarker for patients with NSCLC [110,111], urothelial carcinoma [112], and renal cell cancer [113]. Nevertheless, evaluating PD-L1 presents inherent challenges due to its expression in both neoplastic and non-neoplastic cell populations, significant marker variability, and non-intuitive cutoff values [114]. For instance, Cheng G. et al. demonstrated that DL-based AI diagnostic workflows exhibited high performance in scoring PD-L1 [115]. The research explored and optimized three separate AI model-based workflows to automatically identify positive PD-L1 expression [114]. The AI-supported DL diagnostic models exhibited a notably accurate performance in detecting both lung ADC and lung SqCC across various sampling methods, especially regarding PD-L1 expression at the 1% threshold [114]. These results imply that AI-driven diagnostic models offer potential in assisting pathologists with precise assessments of PD-L1 expression.

#### 3.2.4. Malignant Melanoma

Diagnosing pigmented lesions is one of the most challenging tasks in pathology, requiring extensive training and expertise [116]. Malignant melanoma is among the types of cancer whose incidence and mortality has significantly increased in recent decades [117]. The rise of immunotherapy strategies, particularly immune checkpoint inhibitors (ICIs) directed at PD-1 and its ligand PD-L1, has marked a substantial paradigm shift in the treatment of malignant melanoma, leading to an impressive 58% increase in the three-year median survival rate [118,119,120].

Using WSI and radiomics, AI could help to identify EGFR mutations [121,122], ALK [123], and PD-L1 expression [124,125,126]. For example, in their research, Kearney S. et al. used computational Tissue Analysis (cTA™) to illustrate that, in nearly all instances, the within-sample standard deviation of the cTA™ digital score PD-L1 results was lower than that of the manual score. Specifically, the median inter-pathologist coefficient of variation (%CVs) decreased from 124.9% to 7.8%, and the intra-pathologist coefficient decreased from 65.4% to 7.6%, for manual and digital scores, respectively [127].

Melanoma staging typically involves measuring the proliferative activity of cells, a fundamental process in tumors [127]. The proliferation index (PI), which assesses tumor progression and informs future therapy, is determined by estimating the ratio of active nuclei to total nuclei, using Ki67 stain [128]. In their study, Alheejawi S. et al. used a DL algorithm for the automatic measurement of proliferation index (PI) values in Ki67 stained biopsy images. Experimental results demonstrate that the nuclei segmentation technique using a CNN model, can classify nuclei with low computational complexity, achieving an average error rate of less than 0.7% [128].

#### 3.2.5. Cancers of Unknown Primary Origin (CUPs)

Despite significant efforts, it is currently estimated that in about 3% of metastatic patients, the tissue of origin of the neoplastic lesion remains unidentified, leading to a diagnosis of CUP [129]. Gene expression profiling [130] and epigenetic profiling [131] have been introduced to identify the tissue of origin in metastatic cancers. While these methods have shown promising results, they are not yet widely implemented in clinical practice. Consequently, pathologists continue to rely on standard IHC to determine the tissue of origin [131]. This often involves testing various tissue-specific markers, which can deplete small specimens and frequently fail to resolve the diagnostic issue [132]. Most cases of CUP are carcinomas, which are categorized into well or moderately differentiated adenocarcinomas (60%), undifferentiated or poorly differentiated adenocarcinomas (30%), squamous-cell carcinomas (5%), and undifferentiated neoplasms (5%) [133,134,135].

Due to the constant evolution of antibody repertoires, it is not feasible for pathologists to memorize all molecular marker expressions across different tumors [136]. Algorithmic approaches and standardized IHC panels help but they are time consuming and labor intensive for unique cases [137,138,139]. Therefore, an expert system using computer software (ImmunoGenius version 1.1) was developed by Chong Y. et al. as an iOS and Android application, based on an ML algorithm and an IHC database, to assist pathologists in making precise diagnoses [140]. The precision rates (proportion of correctly identified instances out of the total number of instances) were 78.5%, 78.0%, and 89.0% for the training, validation, and test datasets, respectively, indicating no significant difference between them [140]. The main reason for discordant precision was the lack of disease-specific IHC markers and the overlapping IHC profiles observed in similar diseases [140].

#### 3.2.6. Lymphoid Neoplasms

Lymphomas are a heterogeneous group of malignancies that arise from the clonal proliferation of B-cell, T-cell and natural killer (NK) cell subsets of lymphocytes at different stages of maturation [141,142], and they encompass 5% of all cancers [143]. Further classification is based on the maturation stage, phenotypic character, morphologic features, clinical information, and cytogenetic/molecular genetic findings [144]. When pathologists encounter lymphoma cases, they will examine the H&E slides at both low and high magnifications [145]. Low magnification (4× and 10×) is used to evaluate the overall tissue architecture, determining whether the growth pattern is follicular, which can indicate follicular lymphoma, or diffuse, which may suggest diffuse large B-cell lymphoma (DLBCL) [145]. High magnification (20× and 40×) is utilized to closely inspect cytomorphological features, including nuclear chromatin texture and nucleoli [145]. Although morphological criteria are established for recognizing these diseases, both IHC and genetic tests are necessary to support the diagnosis, and the integration of AI in these tests may simplify the process. El Achi et al. (2019) used a CNN algorithm to differentiate between H&E slides of benign lymph nodes, DLBCL, small lymphocytic lymphoma (SLL), and Burkitt lymphoma across 128 cases and demonstrated high diagnostic accuracy [146].

There are some studies that demonstrate the utility of AI software for the quantification of IHC markers on lymphoma slides. For example, Chong Y. and his colleagues conducted a study in 2020 using ImmunoGenius on 150 cases of lymphoma (H&E and IHC slides), and the diagnostic precision produced acceptable success rates [147]. The results were excellent for most B-cell lymphomas (DLBCL, follicular lymphoma, and SLL, with zero error rates), and the performance for T-cell lymphomas was generally equivalent (T lymphoblastic leukemia/lymphoma and extranodal NK/T-cell lymphoma, nasal type, with zero error rates) [147]. Furthermore, Abdul-Ghafar J. et al. (2023) also studied the utility of Immunogenius and shared the same conclusions as his colleagues regarding lymphomas [148]. In addition, they emphasized that the primary reasons for inaccurate precision were atypical IHC profiles in certain cases, the absence of disease-specific markers, and overlapping IHC profiles among similar diseases [148]. Moreover, in 2020, Carreras J. et al. demonstrated that a high IHC expression of TNFAIP8, determined using an AI-based segmentation method, is associated with a poor overall survival in patients with DLBCL [149].

Table 2 summarizes scientific articles that analyze the use of AI in IHC. 

## 4. Discussion and Challenges

Our literature review highlights the significant potential of AI in IHC, emphasizing its ability to transform diagnostic procedures and enhance patient care by improving accuracy, efficiency, and overall diagnostic outcomes. We found a substantial number of articles focusing on the implementation of AI on H&E slides. However, since the application of AI in medical practice is relatively new, there are still many ongoing studies specifically related to IHC. Our research identified a higher number of studies employing AI algorithms on IHC markers in breast pathology, making it the most extensively studied pathology in this domain. AI algorithms, particularly those based on DL, have shown the ability to analyze complex patterns on IHC slides and subtle differences in tissue samples that might be overlooked by the pathologist. The automation of IHC analysis enhances diagnostic consistency and speeds up the process, a crucial factor when timely diagnosis is essential for initiating appropriate treatment plans.

Available AI tools, such as mobile applications, PC software, and free internet platforms, can now be used by pathologists around the world. These tools can decrease turnaround time and the workload of medical specialists, providing more precise diagnoses to patients. Moreover, software developments should aim to incorporate all IHC markers into a single platform that should be free, accessible, and user-friendly.

Despite these promising advancements, the integration of AI in IHC is not without its challenges. The first challenge is the quantity and quality of training data used to develop the AI algorithm. There are significant variations in the file formats of whole slide images (WSIs), scanner quality, and glass slide quality, including differences in staining intensity, coverslip conditions, tissue size, folded tissue, and the presence of air bubbles, among others [64]. An open-source quality control tool for DP slides, HistoQC, has significantly improved the overall concordance among pathologists in identifying unsuitable WSIs for computational analysis [139]. Another concern for scientists and medical specialists is that AI tools must meet rigorous standards for accuracy, safety, and ethical considerations. Systems and applications with significant and nominal safety implications should be managed according to established protocols. Models for personalized risk estimates must be well calibrated and efficient, with effective updating protocols in place [150].

The authors used a visual representation, depicted in Figure 2, to outline the steps adopted to reduce incorrect cell identification and improve diagnosis.

## 5. Conclusions

In conclusion, the AI algorithms developed thus far have exhibited considerable potential in assisting medical specialists, especially pathologists, in the precise diagnosis of H&E slides and the accurate counting and quantification of IHC markers. Despite the substantial progress, numerous challenges remain, concerning the application and implementation of AI, particularly in countries with low socio-economic status. However, ongoing efforts by specialists and researchers are concentrated on digitizing the diagnostic process, minimizing errors caused by human observation, and optimizing the protocols through which oncologists treat individual patients, thereby bringing precision medicine closer to realization. These initiatives aim to enhance the accuracy and efficiency of cancer diagnosis and treatment, ultimately paving the way for more personalized and effective healthcare solutions.

## Figures and Tables

**Figure 1 jpm-14-00693-f001:**
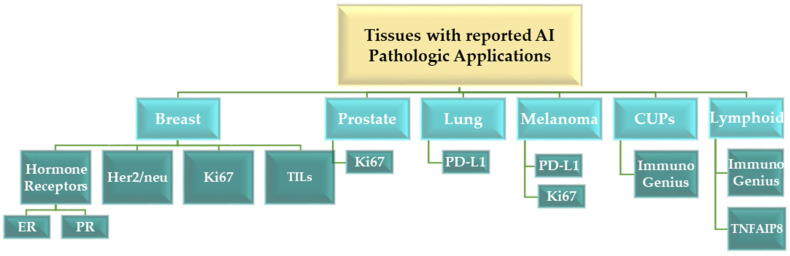
Illustration of the tissues with reported AI pathologic applications (AI—Artificial Intelligence; CUPs—Cancers of Unknown Primary Origin; ER—Estrogen Receptor; PR—Progesterone Receptor; Her2/neu—Human Epidermal Growth Factor Receptor 2; Ki67—proliferation index; TILs—Tumor-Infiltrating Lymphocytes; PD-L1—Programmed Death-Ligand 1; TNFAIP8—Tumor Necrosis Factor Alpha-Induced Protein 8).

**Figure 2 jpm-14-00693-f002:**
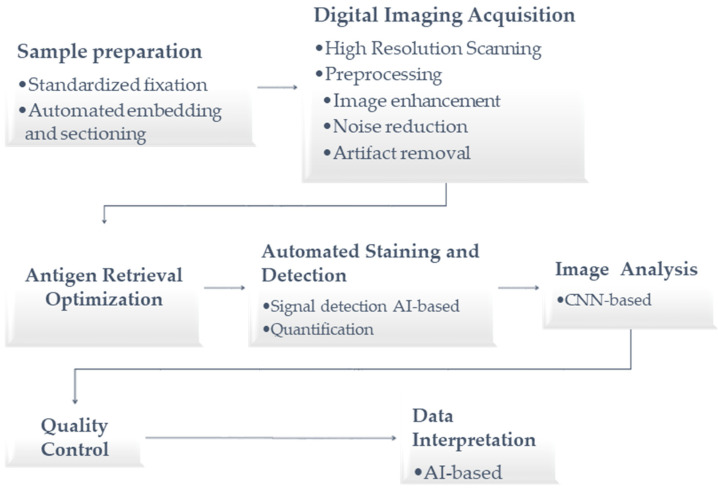
Steps adopted to reduce incorrect cell identification and improve diagnosis.

**Table 1 jpm-14-00693-t001:** Definitions of terms related to the use of AI in pathology.

Term	Definition
Pathology	Pathology (from the Greek word pathología, meaning the study of suffering) refers to the medical specialty focused on the origins, progression, structural and functional alterations, and inherent course of diseases [6]. Modern pathology practice relies primarily on morphology, which encompasses both macroscopic and microscopic features observed on hematoxylin and eosin (H&E)-stained slides, supported by additional tools, such as special stains, immunohistochemical markers, genetics, cytological samples, clinical and radiological evaluations.
Immunohistochemistry (IHC)	IHC is a widely used complementary testing method in pathology for cell classification and diagnosis. It utilizes antibodies targeted against certain antigens in specific tissues and cells to facilitate determination of cell type and organ of origin [7]. Traditional IHC employs a single antibody for each tissue section, and multiplex IHC (mIHC) permits labeling of six or more distinct cell types within a single histologic tissue section [8].
Digital Pathology (DP)	A term that encompasses tools and systems to digitize pathology slides and associated meta-data, their storage, review, and analysis, and enabling infrastructure [9]. DP systems typically WSI scanners to capture high-resolution images of entire glass slides (H&E, IHC, cytology).
Whole Slide Imaging (WSI)	WSI involves 2 procedures: the first process uses specialized hardware to scan glass slides into large digital images; the second process utilizes specialized software to view and analyze these digital files [10,11,12]. A WSI scanner is a device used to convert entire glass microscope slides into high-resolution digital images. All modern WSI systems include illumination systems, microscope optical components, and a focusing system that precisely places an image on a camera [13].
Computational Pathology (CPATH)	CPATH is defined by Louis D.N. et al. (2014) as an approach to diagnosis that incorporates multiple sources of raw data, such as clinical electronic medical records, laboratory data, and imaging. Biologically and clinically relevant information can be extracted, mathematical models are used to generate diagnostic inferences and predictions, and actionable knowledge is presented through integrated reports and interfaces. This enables physicians, patients, laboratory personnel, and other healthcare system stakeholders to make the best possible medical decisions [14].
Machine Learning (ML)	ML, a subset of AI exhibits the experiential “learning” associated with human intelligence, while also having the capacity to improve its analyses using computational algorithms [15]. It can be broadly subclassified in three main categories, such as supervised learning (with specific algorithms, for example linear and logistic regression, neural networks, and decision trees), unsupervised learning, and semi-supervised learning [16]. Supervised learning demonstrates an algorithm’s capability to generalize knowledge from available data with target or labeled cases, enabling the algorithm to predict new (unlabeled) cases [17]. Unsupervised learning involves using algorithms to automatically group unclassified data into clusters based on underlying relationships or features [17]. Semi-supervised learning is a type of ML that utilizes a small amount of labeled data along with a large amount of unlabeled data for training [17].
Decision Tree	A decision tree is a form of supervised ML algorithm structured as a tree, which builds a model to predict the value of a target variable using several input features. eXtreme Gradient Boosting (XGBoost) is a scalable end-to-end tree boosting system, which is used widely by data scientists to achieve state-of-the-art results on many ML challenges [5].
Computer Aided Diagnosis (CAD)	In CAD, ML techniques are employed to analyze both imaging and non-imaging data from past case samples of a patient population, creating a model that correlates the extracted information with specific disease outcomes [18].
Computer Assisted Image Analysis (CAIA)	CAIA involves utilizing computer algorithms and software to examine and interpret medical images. For instance, some tools can be used to determine whether a breast lesion is benign or malignant, identify the cancer type, assist pathologists in this task, and reduce the variability in results due to observer differences [19].
Deep Learning (DL)	DL enables computational models with multiple processing layers to learn data representations across various levels of abstraction. These methods have significantly advanced the state-of-the-art in fields such as speech recognition, visual object recognition, object detection, and other areas including drug discovery and genomics [20].
Convolutional Neural Networks (CNNs)	A CNN is a DL algorithm specifically designed for image and video processing, making it a popular choice for medical image analysis and diagnostics. CNNs are preferred because they are robust and easy to train [21,22,23].
Recurrent Neural Networks (RNNs)	An RNN is a DL algorithm designed to address various interdependent problems simultaneously. It features a structure organized in closed loops, enabling the network to effectively handle the interdependencies of tasks [24,25].
Generative Adversarial Networks (GANs)	A GAN is a DL algorithm in which, during training, information from images is integrated with a statistical predictor, collaboratively determining the outcomes [26].

**Table 2 jpm-14-00693-t002:** Scientific articles that analyze the use of AI in IHC.

Pathology	Year of Study	Author	IHC Application	Key Findings of the Reported Studies
Breast Cancer	2020	Rawat R. R. [56]	ER, PR, HER2	Prediction of the clinical subtypes of breast cancer
	2020	Saha M. [57]	ER, PR	Scoring and quantification
	2022	Shafi S. [59]	ER	Quantification and comparison to manual assessment
	2016	Helin H.O. [66]	HER2	Evaluation and comparison
	2015	Holten-Rossing H. [67]	HER2	Automated reading and comparison to conventional manual assessment
	2022	Hartage R. [68]	HER2	Analysis and correlation with HER2/neu-FISH
	2022	Li L. [39]	Ki67	Quantification and comparison to manual assessment
	2023	Abele N. [71]	ER, PR, Ki67	Quantification and analysis
	2023	Erber R. [72]	Ki67	Automated reading and comparison to conventional manual assessment
	2018	Swiderska-Chadaj Z. [81]	TILs	Detection and quantification
	2014	Chen T. [83]	TILs	Automatic immune cell counting
Prostate Cancer	2023	Blessin N.C. [98]	Ki67	Automated assessment
Lung Cancer	2022	Cheng G. [115]	PD-L1	Evaluation and comparison to manual assessment
Malignant Melanoma	2017	Kearney S. [127]	PD-L1	Digital scoring and comparison with manual assessment
	2019	Alheejawi S. [128]	Ki67	Automated quantification
Cancer of Unknown Primary Origin	2021	Chong Y. [140]	ImmunoGenius	Diagnosis prediction on IHC database
Lymphoid neoplasms	2020	Chong Y. [147]	ImmunoGenius	Diagnosis prediction on lymphomas
	2023	Abdul-Ghafar J. [148]	Immunogenius	Diagnosis prediction on lymphomas
	2020	Carreras J. [149]	TNFAIP8	Prognosis prediction

## Data Availability

Not applicable.

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
