# Peer review of "Revolutionizing Pathology with Artificial Intelligence: Innovations in Immunohistochemistry"

_jpm, 2024, doi:10.3390/jpm14070693_

Round 1
Reviewer 1 Report
Comments and Suggestions for Authors
Dear Authors,
The paper is conceptually very interesting and important from a practical point of view. The article examines artificial intelligence (AI) and successfully implemented it in pathology.
But, research indicates that the automated approach for marking tumor cells is not as precise as the visible method.
AI may identify some tumor cells incorrectly, such as misidentifying interstitial cells as tumor cells or ignoring positive tumor cells with fuzzy outlines and lighter staining, due to the heterogeneity of cancer cells.
Could you give a graphic summary of the steps used to decrease incorrect cell identification and enhance diagnosis?
----
Additional comment:
Manuscript jpm-3078988 uses approximately 45% pre-existing text. While passages citing personal data and methods are acceptable, there are significant sections of the main content that use pre-existing text without any interpretation or analysis. I recommend revising these sections to reduce the use of pre-existing text to 20%.
Author Response
Thank you for giving us the opportunity to submit a revised draft of the manuscript “Revolutionizing Pathology with Artificial Intelligence: Innovations in Immunohistochemistry” for publication in the Journal of Personalized Medicine. We appreciate the time and effort that you dedicated to providing feedback on our manuscript and are grateful for the insightful comments. We have incorporated your suggestions. Those changes are highlighted in red within the manuscript. Please see below, for a point-by-point response to your comments and concerns.
Point 1: The paper is conceptually very interesting and important from a practical point of view. The article examines artificial intelligence (AI) and successfully implemented it in pathology. But, research indicates that the automated approach for marking tumor cells is not as precise as the visible method.
AI may identify some tumor cells incorrectly, such as misidentifying interstitial cells as tumor cells or ignoring positive tumor cells with fuzzy outlines and lighter staining, due to the heterogeneity of cancer cells.
Could you give a graphic summary of the steps used to decrease incorrect cell identification and enhance diagnosis?
Response 1: Thank you for your feedback and your valuable suggestion. In response, we have created a graphic summary, presented in Section 4: Discussions and challenges, which outlines the methods employed to reduce incorrect cell identification and improve diagnostic accuracy.
Point 2: Manuscript jpm-3078988 uses approximately 45% pre-existing text. While passages citing personal data and methods are acceptable, there are significant sections of the main content that use pre-existing text without any interpretation or analysis. I recommend revising these sections to reduce the use of pre-existing text to 20%.
Response 2: Thank you for your detailed review of our manuscript. We appreciate your insights and recognize the concerns regarding our document's high percentage of pre-existing text. We thoroughly reviewed the manuscript to identify all instances of pre-existing text and aimed to reduce its use as recommended. Where the use of pre-existing text was unavoidable, we ensured it was properly paraphrased and cited according to the journal's guidelines.

Reviewer 2 Report
Comments and Suggestions for Authors
A manuscript by Poalelungi et al. summarizes the role of artificial intelligence (AI) related to immunohistochemistry in revolutionizing pathology. The paper has clear rationale and methodology. Data acquisition is well described. However, the presentation of data needs to be further improved. The authors listed breast, prostate, lungs, and melanocytes as targets of AI-related immunohistochemical studies. This manuscript would benefit from adding drawings to these tissues showing the cellular and subcellular localization of the investigated factors. Inclusion of these drawings will enhance the readability of this work, especially by junior scientists and beginners to the field of pathology.
Minor comments are listed below:
Abstract
The abstract is nicely written. However, long sentences are distracting. Authors are advised to use shorter sentences with sharp and clear meanings. Both the problem statement and study conclusion should be enhanced.
Introduction
Line 29: tissues samples change to “tissue samples”.
Line 30: widely-used, remove the hyphen (widely used).
Line 34: Digital Pathology change to “Digital pathology” or keep as it is and abbreviate as (DP).
Table 1. contains important definitions, though the word casing should be thoroughly revised and ensured to be consistent throughout the table.
Table 1. Whole Slide Imaging (WSI). WSI involves two processes: first process implies using specialized hardware (scanner) to digitize glass slides into large digital images; second, utilizing specialized software to view and analyze these extensive digital files [10-12]. Please review the grammar of this part.
Table 1. Computational Pathology (CPATH). Write this part in the passive tense and keep the citation at the end.
Table 1. Machine Learning (ML). Briefly define the unsupervised, and the semi-supervised learning.
Figure 1. Top row should be changed to ” Tissues with reported AI Pathologic Applications”. The figure title should be changed accordingly. Add few explanations to the third level rows of the figure, for instance: Hormone receptors (ER, PR) and so on. All abbreviations on the figure should be listed in full in the figure legend. Correct the word “lymphocites”.
2. Role of Artificial Intelligence in Immunohistochemistry
Line 67: Fibroblast growth factor receptor-2 → fibroblast growth factor receptor-2
Line 74: 2.2. Integration with Digital Pathology. Integrating what with Digital Pathology? Please clear the meaning.
3. Literature Review
Line 149: The evaluation of hormone receptors (HR) status → The evaluation of hormone receptor (HR) status.
Line 220: Paige Prostate is a ML algorithm → Paige Prostate is an ML algorithm or Paige Prostate is a machine learning algorithm.
Line 291,292: intra-pathologists from 65.4% to 7.6% → intra-pathologists decreased from 65.4% to 7.6%
Table 2. add a column summarizing the main findings of the reported studies.
Line 332: H&E slides.
Author Response
Thank you for giving us the opportunity to submit a revised draft of the manuscript “Revolutionizing Pathology with Artificial Intelligence: Innovations in Immunohistochemistry” for publication in the Journal of Personalized Medicine. We appreciate the time and effort that you dedicated to providing feedback on our manuscript and are grateful for the insightful comments. We have incorporated your suggestions. Those changes are highlighted with yellow within the manuscript. Please see below, for a point-by-point response to your comments and concerns.
Point 1: Abstract
The abstract is nicely written. However, long sentences are distracting. Authors are advised to use shorter sentences with sharp and clear meanings. Both the problem statement and study conclusion should be enhanced.
Response 1: Thank you for your kind words! We have shortened the sentences and slightly expanded the problem statement and study conclusion.
Point 2: Introduction
Response 2: Thank you for pointing all those faults!
Line 29: tissues samples change to “tissue samples”.
Now Line 32: we changed to “tissue samples”.
Line 30: widely-used, remove the hyphen (widely used).
Now Line 33: we removed the hyphen.
Line 34: Digital Pathology change to “Digital pathology” or keep as it is and abbreviate as (DP).
Now Line 37: We kept Digital Pathology and we abbreviated as (DP).
Table 1. contains important definitions, though the word casing should be thoroughly revised and ensured to be consistent throughout the table.
We replaced the casing of words with their abbreviations in the definitions (highlighted).
Table 1. Whole Slide Imaging (WSI). WSI involves two processes: first process implies using specialized hardware (scanner) to digitize glass slides into large digital images; second, utilizing specialized software to view and analyse these extensive digital files [10-12]. Please review the grammar of this part.
We corrected as: WSI involves 2 procedures: the first process uses specialized hardware to scan glass slides into large digital images; the second process utilizes specialized software to view and analyse these digital files.
Table 1. Computational Pathology (CPATH). Write this part in the passive tense and keep the citation at the end.
We wrote using passive constructions and kept the citation.
Table 1. Machine Learning (ML). Briefly define the unsupervised, and the semi-supervised learning.
We added definitions for Supervised, Unsupervised and Semi-supervised Learning.
Figure 1. Top row should be changed to ”Tissues with reported AI Pathologic Applications”. The figure title should be changed accordingly. Add few explanations to the third level rows of the figure, for instance: Hormone receptors (ER, PR) and so on. All abbreviations on the figure should be listed in full in the figure legend. Correct the word “lymphocites”.
We changed the top row’s name and the figure’s title. We added ER, PR and the legend, and corrected the word lymphocytes.
Point 3: 2. Role of Artificial Intelligence in Immunohistochemistry
Line 67: Fibroblast growth factor receptor-2 → fibroblast growth factor receptor-2
Now Line 76 - fibroblast growth factor receptor-2
Line 74: 2.2. Integration with Digital Pathology. Integrating what with Digital Pathology? Please clear the meaning.
We changed with “Integration of AI Software with Digital Pathology”
Point 4: 3. Literature Review
Line 149: The evaluation of hormone receptors (HR) status → The evaluation of hormone receptor (HR) status.
Now Line 158 – we corrected The evaluation of hormone receptor (HR) status.
Line 220: Paige Prostate is a ML algorithm → Paige Prostate is an ML algorithm or Paige Prostate is a machine learning algorithm.
Now Line 229 - Paige Prostate is a machine learning algorithm.
Line 291,292: intra-pathologists from 65.4% to 7.6% → intra-pathologists decreased from 65.4% to 7.6%
Now Line 300-301: intra-pathologists decreased from 65.4% to 7.6%.
Table 2. add a column summarizing the main findings of the reported studies.
We added a column in table 2.
Line 332: H&E slides.
Now Line 375: H&E slides.

Reviewer 3 Report
Comments and Suggestions for Authors
The manuscript provides an extensive review of the current state of AI in IHC, covering various aspects of its implementation and highlighting its potential in transforming diagnostic procedures. It discusses on breast pathology as the most extensively studied area is well-supported and relevant, given the prevalence and significance of breast cancer. The manuscript outlines how AI can increase patient care by improving diagnostic accuracy, efficiency, and overall outcomes. The manuscript presents a thorough and well-researched review of the potential of AI in IHC, highlighting both its advantages and challenges.
Including specific case studies where AI has successfully been applied in IHC would strengthen the manuscript. The discussion on ethical considerations could be expanded to include potential biases in AI algorithms and strategies to lessen these issues.
Specific comments:
1. Line 336: Define "DL"
2. Lines 355-357: The conclusion could be strengthened by emphasizing the future directions of research and implementation.
With these revisions the manuscript may have improved detail and readability.
Author Response
Thank you for giving us the opportunity to submit a revised draft of the manuscript “Revolutionizing Pathology with Artificial Intelligence: Innovations in Immunohistochemistry” for publication in the Journal of Personalized Medicine. We appreciate the time and effort that you dedicated to providing feedback on our manuscript and are grateful for the insightful comments. We have incorporated your suggestions. Those changes are highlighted with green within the manuscript. Please see below, for a point-by-point response to your comments and concerns.
Point 1: The manuscript provides an extensive review of the current state of AI in IHC, covering various aspects of its implementation and highlighting its potential in transforming diagnostic procedures. It discusses on breast pathology as the most extensively studied area is well-supported and relevant, given the prevalence and significance of breast cancer. The manuscript outlines how AI can increase patient care by improving diagnostic accuracy, efficiency, and overall outcomes. The manuscript presents a thorough and well-researched review of the potential of AI in IHC, highlighting both its advantages and challenges.
Including specific case studies where AI has successfully been applied in IHC would strengthen the manuscript. The discussion on ethical considerations could be expanded to include potential biases in AI algorithms and strategies to lessen these issues.
Response 1: Thank you for your kind words! We included some of the specific case studies (e.g. TILs, Ki67, Immunogenius)
Point 2: Line 336: Define "DL"
Response 2: DL comes from Deep Learning, first defined and abbreviated in Table 1.
Point 3: Lines 355-357: The conclusion could be strengthened by emphasizing the future directions of research and implementation.
Response 3: We appreciate the reviewer’s feedback! We added a small paragraph about future directions.

Reviewer 4 Report
Comments and Suggestions for Authors
In the manuscript titled "Revolutionizing Pathology with Artificial Intelligence: Innovations in Immunohistochemistry," a team led by Poalelungi and Fulga describes the various applications and assesses the present status of Artificial Intelligence (AI) integration in immunohistochemical analysis.
The manuscript presents several interesting points in a well-structured manner. However, some minor weaknesses need to be addressed before it can be published.
1. In addition to the AI application in solid tumors, hematological tumors should also be included in this manuscript. Relevant PMID numbers such as 38587646, 38452316, and 34612160 could be useful.
2. The application of AI in IHC in colon cancer should also be included in this manuscript. Relevant PMID numbers such as 33888518 and 36284712 could be useful.
3. Could the author introduce the application of AI-based tools in pathology?
Author Response
Thank you for allowing us to submit a revised draft of the manuscript “Revolutionizing Pathology with Artificial Intelligence: Innovations in Immunohistochemistry” for publication in the Journal of Personalized Medicine. We appreciate the time and effort that you dedicated to providing feedback on our manuscript and are grateful for the insightful comments. We have incorporated your suggestions. Those changes are highlighted in blue within the manuscript. Please see below, for a point-by-point response to your comments and concerns.
Point 1: In the manuscript titled "Revolutionizing Pathology with Artificial Intelligence: Innovations in Immunohistochemistry," a team led by Poalelungi and Fulga describes the various applications and assesses the present status of Artificial Intelligence (AI) integration in immunohistochemical analysis.
The manuscript presents several interesting points in a well-structured manner. However, some minor weaknesses need to be addressed before it can be published.
Response 1: Thank you for your kind words!
Point 2: In addition to the AI application in solid tumors, hematological tumors should also be included in this manuscript. Relevant PMID numbers such as 38587646, 38452316, and 34612160 could be useful.
Response 2: Thank you for those recommendations! We included 3.2.6. Lymphoid neoplasms with citations 141-149. However, PMID 38587646 is about radiomics and PET-CT correlation and does not discuss immunohistochemistry. PMID 38452316 does not include any immunohistochemical markers, focusing solely on cytogenetics with fluorescence in situ hybridization (FISH). Lastly, PMID 34612160 does not meet the criteria outlined in our study.
Point 3: The application of AI in IHC in colon cancer should also be included in this manuscript. Relevant PMID numbers such as 33888518 and 36284712 could be useful.
Response 3: We appreciate the reviewer’s feedback! To the best of our knowledge, and after extensive research, we found no studies suitable for inclusion in our analysis of IHC marker quantification in colorectal cancer. While some studies incorporate H&E images, none specifically focus on IHC. Additionally, the studies recommended (PMID 33888518 and 36284712) employ methods other than IHC.
Point 4: Could the author introduce the application of AI-based tools in pathology?
Response 4: The article mentions those applications involving IHC which are schematized in Figure 1.
